# Added Value of Scintillating Element in Cerenkov-Induced Photodynamic Therapy

**DOI:** 10.3390/ph16020143

**Published:** 2023-01-18

**Authors:** Perrine Schneller, Charlotte Collet, Quentin Been, Paul Rocchi, François Lux, Olivier Tillement, Muriel Barberi-Heyob, Hervé Schohn, Joël Daouk

**Affiliations:** 1Department of Biology, Signals and Systems in Cancer and Neuroscience, UMR 7039, Université de Lorraine, French National Scientific Research Center (CNRS), Research Center for Automatic Control (CRAN), UMR CNRS 7039 CRAN, Campus Biologie Santé, 9 Avenue de la Forêt de Haye, BP10299, CEDEX, 54505 Vandoeuvre-lès-Nancy, France; 2NancyCloTEP, Molecular Imaging Platform, Université de Lorraine, Brabois Hospital, 54600 Vandoeuvre-lès-Nancy, France; 3IADI, INSERM U1254, Université de Lorraine, 54000 Nancy, France; 4Light Matter Institute, UMR-5306, Université de Lyon, French National Scientific Research Center (CNRS), 69000 Lyon, France; 5NH TherAguix SA, 38240 Meylan, France; 6Institut Universitaire de France (IUF), 75005 Paris, France

**Keywords:** glioblastoma, AGuIX, Terbium, Gadolinium, photodynamic therapy, Cerenkov radiation, singlet oxygen, Monte Carlo simulations

## Abstract

Cerenkov-induced photodynamic therapy (CR-PDT) with the use of Gallium-68 (^68^Ga) as an unsealed radioactive source has been proposed as an alternative strategy to X-ray-induced photodynamic therapy (X-PDT). This new strategy still aims to produce a photodynamic effect with the use of nanoparticles, namely, AGuIX. Recently, we replaced Gd from the AGuIX@ platform with Terbium (Tb) as a nanoscintillator and added 5-(4-carboxyphenyl succinimide ester)-10,15,20-triphenylporphyrin (P1) as a photosensitizer (referred to as AGuIX@Tb-P1). Although Cerenkov luminescence from ^68^Ga positrons is involved in nanoscintillator and photosensitizer activation, the cytotoxic effect obtained by PDT remains controversial. Herein, we tested whether free ^68^Ga could substitute X-rays of X-PDT to obtain a cytotoxic phototherapeutic effect. Results were compared with those obtained with AGuIX@Gd-P1 nanoparticles. We showed, by Monte Carlo simulations, the contribution of Tb scintillation in P1 activation by an energy transfer between Tb and P1 after Cerenkov radiation, compared to the Gd-based nanoparticles. We confirmed the involvement of the type II PDT reaction during ^68^Ga-mediated Cerenkov luminescence, id est, the transfer of photon to AGuIX@Tb-P1 which, in turn, generated P1-mediated singlet oxygen. The effect of ^68^Ga on cell survival was studied by clonogenic assays using human glioblastoma U-251 MG cells. Exposure of pre-treated cells with AGuIX@Tb-P1 to ^68^Ga resulted in the decrease in cell clone formation, unlike AGuIX@Gd-P1. We conclude that CR-PDT could be an alternative of X-PDT.

## 1. Introduction

Glioblastoma multiforme (GBM) is one of the main aggressive brain tumours with poor patient survival [1,2]. The conventional treatment of GBM tumours consists of surgical resection followed by X-ray radiation and adjuvant temozolomide administration, which modestly improves patient survival [3]. Within the tumour, cell exposure to X-ray involves DNA breaks followed by mitotic catastrophe and induced-senescence. Moreover, X-ray exposition also leads to the generation of oxidative stress, and cell redox status changes, triggering cells death [4]. However, these effects are not limited to malignant cells, but also alter surrounding cells.

Photodynamic therapy (PDT) and X-ray-induced phototherapy (X-PDT) are alternative therapeutic strategies to radiotherapy. PDT is based on the use of a photoactivable molecule as a photosensitizer (PS), such as porfimer sodium (Photofrin), 5-aminolevulinic acid (5-ALA, Gliolan), m-tetrahydroxyphenylchlorin (mTHPC, temoporfin, Foscan) and benzoporphyrin derivative monoacids ring A (BPD-MA, verteporfin, Visudine). After visible light excitation and in the presence of oxygen, the PS generates singlet oxygen and reactive oxygen-derived species; the singlet oxygen being the main mediator of the PDT reaction [5,6]. Improving PDT efficiency was achieved by coupling PS to nanoparticles. Of interest, the nanoparticle could be modified by functional moieties to increase the recognition of cell targets with antibodies or peptides, which recognize tumour cell membrane receptors. Such nanoparticle design has largely been described elsewhere [7]. In addition, nanoparticles accumulate in the solid tumour site by the enhanced permeability and retention effect (EPR) [8,9]. Alternative methods, such as interstitial PDT and intraoperative PDT, have also been developed to improve GBM treatment. Particularly, interstitial PDT consists in a local treatment approach giving significant enhanced survival to patients [10,11]. Compared to radiotherapy, the light irradiation is less energetic, limiting the penetration into tissues. Moreover, the therapeutic effect is also counteracted due to the presence of endogen chromophores, which absorb visible light commonly used in clinical practice [5]. Therefore, the penetration depth of 630 nm light in brain-adjacent-to-tumour is estimated at 2.5 mm. This depth penetration can be further improved when PS activation is done in the near infra-red domain [12]. Different strategies can be followed to excite in the near infra-red domain such as the design of up-conversion nanoparticles or two-photons absorption molecules. Nevertheless, this requires the design of complex chemical structures.

X-PDT was developed in order to combine principles of radiotherapy and PDT, both clinically proven modalities; and to maintain their respective benefits while exploiting some synergetic effects. X-PDT proof-of-concept with nanoparticles was first described by Chen and Zhang, who demonstrated simultaneous radiation and X-ray-induced photodynamic effects against tumour cells [13]. Briefly, the X-PDT is based on the conversion of X-ray photons into visible photons from a nanoscintillator embedded in the nanoparticle, but localized near the PS, which produces singlet oxygen after nanoscintillator-mediated energy transduction to a PS. Especially, the nanoscintillator requires a high scintillation quantum yield and an optimal energy transfer from the scintillator onto the photosensitizer [13,14]. However, X-ray energy applied for efficient X-PDT is lower than clinical external radiotherapy beams, limiting treatment efficacy, which could not be observed in a clinical context, limiting further translation.

Recently, another therapeutic strategy was proposed through Cerenkov radiation, referred as Cerenkov-induced PDT (CR-PDT) [15,16]. Cerenkov light is a luminescence signal produced by charged particles such as radionuclides. Two conditions must be present to enable any effect: firstly, the medium must be dielectric and, secondly, the charged particle must travel faster than the phase velocity of light in the medium. In these conditions, Cerenkov photons are produced along the particle path, yielding constructive interferences during successive medium polarization/depolarization [17]. Ran et al. were the first to introduce this strategy and propose that the deleterious impact on cell growth consists of radiotherapeutic effect in synergy with phototherapeutic effect. The latter involved solely PS light-mediated activation [18]. However, the generation of singlet oxygen is dramatically low, since the number of photons absorbed by the PS is lower than those necessary for conventional PDT.

There are only a few original studies with detailed methods of the use of CR-PDT either in vitro or in vivo. Collectively [19,20,21,22,23,24], cancer cell death is increased when a PS and β emitters are associated in the same treatment protocol and the effects observed depend both on the radioactivity deposit and PS concentration absorbed by tumour cells and the ratio of absorbed Cerenkov radiation is enhanced with a higher PS cellular uptake [23]. Particularly, when colocalization of both radio-emitters and the PS is close, an efficient therapeutic effect was observed. It also appears that the photodynamic effect obtained depends on the cell line tested. Such difference in cell line sensitivity involves cell radionuclide accumulation [25]. In addition, the cell lines tested are less or more sensitive to the ionizing radiations and Cerenkov light, suggesting the decrease in CR-PDT cytotoxicity to the overall therapeutic effect. Finally, a real benefit of this therapeutic strategy was demonstrated by reducing tumour growth of tumours developed from grafted cells in nude mice or by the decrease in cancer circulating cells [21,24].

Despite the studies supporting the concept of CR-PDT, there is still a debate about the presence of a cytotoxic phototherapeutic effect [26,27]. We previously performed a theoretical analysis to evaluate the amount of singlet oxygen produced by Cerenkov radiation for diverse PS and radionuclides and we concluded that an efficient PDT treatment could occur, even at low PS concentrations [28]. Herein, we propose to assess the interest of a hybrid nanoplatform AGuIX-like nanoparticle displaying Terbium (Tb) instead of Gadolinium (Gd) in the chelates, as a nanoscintillator, and 5-(4-carboxyphenyl succinimide ester)-10,15,20-triphenylporphyrin (P1), as a PS, and to activate this system by free ^68^Gallium-citrate (^68^Ga) as a positron emitter. Thus, the main objective of the work was to assess whether the replacement of X-ray by ^68^Ga could lead to GBM cell growth arrest with a less radionuclide energy transfer to nanoscintillator. As a consequence, such a therapeutic strategy could be relevant to protect the parenchyma surrounding the tumour against high dose of X-ray ionisation.

To demonstrate the contribution of the scintillator in the CR-PDT, we performed both Monte Carlo simulations and biological assays. Our hypothesis is based on the potential activation of the PS in three distinct ways (Figure 1): (i) From visible photons emitted by ^68^Ga-mediated Cerenkov radiations which will directly excite the PS; (ii) the ultraviolet (UV) part of the Cerenkov spectrum can be absorbed by the scintillating element which, in turn, will transfer this energy to the PS; and (iii) at the end of the free path, positrons will interact with a local electron to produce, after their annihilation, two coincident 511 keV γ-rays. These high energy photons can be absorbed by the scintillator that converted the energy into visible light following the activation of the PS to produce the phototherapeutic effect.

## 2. Results

### 2.1. Singlet Oxygen Production with ^68^Ga Irradiation

We demonstrated singlet oxygen production using Singlet Oxygen Sensor Green (SOSG) fluorescent probe after addition of ^68^Ga in the reaction mixture containing AGuIX@Tb-P1. SOSG signals increased continuously during ^68^Ga exposure (Figure 1). Addition of sodium azide (NaN_3_) or bovine serum albumin (BSA), both singlet oxygen quenchers, confirmed that the type II-PDT reaction (singlet oxygen generation) was mainly involved in CR-PDT as SOSG fluorescence signal increase was almost inhibited. In addition, the singlet oxygen production dynamics followed the ^68^Ga decay as their slope absolute values were similar after logarithmic transformation (Figure 2). Indeed, slope values were 0.0119 and 0.0103 for AGuIX@Tb-P1 and ^68^Ga, respectively, yield a 15% difference between these two functions. This difference can be attributed to measure inaccuracies in experimental data.

### 2.2. Monte Carlo Simulations

Monte Carlo simulations were used to highlight the various physical phenomena and interactions occurring during ^68^Ga decay resulting in PS activation. Different simulations were used to assess the respective Tb scintillation and direct PS activation contribution to the PDT process. Simulation results were obtained on Gate v. 9.1 after 10^6^ positron emission for each scenario.

Cerenkov luminescence was emitted all along the positrons’ tracks in water medium. This luminescence defined a cone as expected (Figure 3a). The corresponding spectrum in water was recorded between 250 and 800 nm (Figure 3b) and Cerenkov luminescence intensity was found inversely proportional to the square of the wavelength.

As optical energy transfer was activated in our simulation, we were able to assess the Cerenkov photons absorbed by P1 given its absorption spectrum. Direct P1 activation by Cerenkov spectrum was observed via the P1 fluorescence peak obtained in absence of lanthanide (Figure 4).

After positron annihilation, 511 keV γ-rays are absorbed by Tb which convert this energy into visible photons through the scintillation physics process. Tb-simulated scintillation was consistent with experimental results as it presented the four typical Tb emission peaks (Appendix A). In addition, Tb absorbs UV photons. Then, due to optical absorption from Tb to the PS, AGuIX@Tb luminescence decreased when P1 was added to the nanoparticle (Figure 4a). The biggest energy transfer was found to involve the 545 nm Tb peak (involving 5D4 → 7F5 transition), highlighted by a three to four time decrease in this scintillation peak. Indeed, the 545 nm Tb peak presents the biggest orbital overlap with P1 Q bands (Appendix A). P1 fluorescence increased twice compared with direct Cerenkov activation. Simulation results were consistent with previous experimental photophysical studies (Appendix A). In addition, Cerenkov photons with wavelength lower than 400 nm can also be absorbed by Tb to produce fluorescence emission at the same wavelengths, similarly to the scintillation process, strengthening the PS activation (Appendix A). To control that P1 fluorescence increase was related to Tb scintillation and fluorescence, we substituted Tb with Gd, which has its emission peak around 300 nm. At this wavelength, Gd cannot excite P1. In this condition, we did not observe any increase in the P1 fluorescence at 650 nm (Figure 4b). Thus, the simulations demonstrated that direct activation of the PS by Cerenkov radiation can be dramatically improved when a suitable scintillator is present in the close vicinity of the PS.

### 2.3. Cerenkov-Induced Photodynamic Effect on U-251 MG Cell Survival

To experimentally validate the benefit of Tb in the vicinity of P1, we studied whether ^68^Ga irradiation promoted U-251 MG cell growth arrest by photodynamic-mediated effect. We performed anchorage-dependent clonogenic assays with AGuIX nanoparticles. Because Gd and Tb are neighbours in terms of atomic number (64 and 65, respectively), their behaviour regarding X-ray interaction is considered as similar. Cells were pre-treated with 1 µM AGuIX@Gd-P1, AGuIX@Tb-P1 or AGuIX@Gd and AGuIX@Tb (for the latter, the concentration used was equivalent to 1 µM P1) and irradiated with increasing ^68^Ga activity concentrations.

Without any prior nanoparticle treatment, we did not find any change of clones’ number after cell exposure to increased ^68^Ga deposition (Figure 5a), suggesting the absence of a ^68^Ga radiotoxicity. Pre-treated cells with AGuIX@Tb-P1 and irradiated in the presence of 1 and 3 MBq of ^68^Ga, led to a significant decrease in the formation of cell clones compared to non-irradiated cells (*p* = 0.004 and *p* = 0.0006, respectively; Figure 5b). Furthermore, compared to the radioactive concentration of 3 MBq, similar results for clonogenic capacity were obtained with cells treated in the presence of AGuIX@Tb-P1 and X-ray irradiated at 2.0 Gy [29]. Moreover, we did not observe any effect on clone formation for activities less than 1 MBq. Similarly, when U-251 MG cells were exposed to AGuIX@Tb without P1, we did not observe any decrease in the number of clones (Figure 5b). A similar experiment was performed with AGuIX-like nanoparticles with Gd with or without P1, AGuIX@Gd or AGuIX@Gd-P1 (Figure 5c). The main difference between them comes from the scintillation/fluorescence (in the visible domain) of Tb capabilities compared to Gd. Hence, we expected a lower PDT efficacy with AGuIX@Gd-P1 treatment. We did not find any difference in U-251 MG clones obtained after exposure to AGuIX@Gd-P1 whatever ^68^Ga activity used (Figure 5c), supporting the absence of radiosensitization of the nanoparticles in the experimental conditions tested. Indeed, AGuIX@ nanoplatforms were reported to have a radiosensitizing effect, which depends on the internalised nanoparticle concentration within the cells [30]. Finally, the results strengthen the simulation results (Figure 3b and Figure 4), supporting the benefit of using Tb instead of Gd in the AGuIX@ platform design to perform CR-PDT.

## 3. Discussion

### 3.1. Scintillator Luminescence Increases Porphyrin Excitation

Cerenkov radiation was modelled in Geant4-Gate in a realistic way by introducing, on the one hand, the kinetic energy of the charged particle and, on the other hand, the angular distribution of the Cerenkov photons by taking into account the refractive index on the medium and the Cerenkov energy [31]. We observed that Cerenkov photons were emitted in a forward direction as expected given the positron mean step length used in our study [32].

Our simulations demonstrated the energy transfer from Cerenkov photons emitted along positron range and the PS (Figure 4). Indeed, single P1 exposed to ^68^Ga highlighted its characteristic fluorescence peak at 652 nm. When we modelled the AGuIX@Gd-P1 nanoparticle, we did not observe any P1 fluorescence increase as expected as there is no overlap between Gd luminescence (~350 nm) and P1 absorption spectrum, id est, Soret band at 405 nm [29]. AGuIX@Tb exposed to ^68^Ga solution showed the characteristic Tb scintillation spectrum. When we added P1 to the nanoparticle, P1 fluorescence increased twice as Tb peaks decreased, demonstrating the energy transfer between Tb and P1. The same observation was made experimentally in a previous work dealing with the same nanoplatform under X-ray exposure [29]. Our simulations revealed that scintillation PDT can be dramatically improved when scintillators are added to the PS.

Indeed, primary positron emission yields Cerenkov emission which will interact with both the PS and the nanoscintillator. However, many other Cerenkov photons are also produced after positron annihilation: The energy from 511 keV γ-rays, absorbed by Tb or Gd by photoelectric effect, induce photoelectron emission with a high kinetic energy (e.g., 511 keV minus electron binding energy and potential energy loss by prior Compton scattering). These photoelectrons will, in turn, generate a Cerenkov effect along their paths (~1 mm in biological medium). In such conditions, there is no need for the isotope carrier to be highly specific to the targeted tumour. Then, grafting the radioisotope onto a separate drug other than the PS should be the optimal strategy: the PS should be linked onto a nanocarrier highly specific to the tumour to reduce potential side effects to healthy tissues. On a second occasion (with respect to the drug-light delay), the radiotracer, which is specific to the tumour and its vicinity, would be injected to create a radioactive volume encompassing the tumour, its vasculature and surrounding tissues benefiting from all the different ways that Cerenkov light is generated.

### 3.2. Low activity Concentration Yields Singlet Oxygen Production

As shown in Figure 1a, singlet oxygen can be produced when PSs are in presence of β emitters. We also demonstrated that this production dynamic followed the isotope decay rule (Figure 1b). Then, the optimal radiotracer would be a highly specific molecule to the tumour and its neighbourhood labelled with a high energy isotope with a long half-life. Indeed, following the Frank–Tamm rule, the lowest (positron or electron) initial kinetic energy to be effective in biological tissues is around 250 keV [33]. We used SOSG fluorescent probe in our experiments [34,35]. Although being specific to singlet oxygen, some reactions inducing endoperoxide derivative have been reported under UV excitation [36]. With Cerenkov radiations also emitting UV light, potential bias would have flawed our results. Nevertheless, when we exposed AGuIX@Tb or AGuIX@Gd to high energy photons, we did not observe any SOSG increase at the activity concentrations used in our previous experiments [29].

### 3.3. ^68^Ga Irradiation of AGuIX@Tb-P1 Induces an Effective Photodynamic Effect

As shown in Figure 1 and Figure 2, addition of free ^68^Ga with AGuIX@Tb-P1 led to the generation of singlet oxygen, the main product of photodynamic effect. Moreover, nanoparticles’ pre-treated cell exposure to ^68^Ga is associated with the decrease in the capability of cell clone formation (Figure 5). Cell treatment conditions, as 1 µM equivalent Tb, were previously chosen since cell clone formation was mainly inhibited when cells were pre-treated with 2.5 µM AGuIX@Tb-P1 and then irradiated with a low X-ray dose [29]. Interestingly, when cells were exposed to AGuIX@Gd, AGuIX@Tb or AGuIX@Gd-P1, cell growth was not significantly lowered (Figure 5b,c) supporting the concept that both Tb, as NS, and P1, as PS, are needed to obtain an efficient therapeutic effect related to the phototherapeutic performance by positron and Cerenkov radiations. However, it must be pointed out that radiosensitizing effects on cells has been reported for the original AGuIX@ platform containing Gd [30,37]. The Gd concentration to obtain such radiosensitive impact on cancer cells was reported for nanoparticles concentrations from 100 µM up to 1 mM [38]. Herein, the concentration of lanthanide was at least four times lower than the minimal concentration reporting effective radiosensitization, respectively, 16 and 25 µM for Tb and Gd (Figure 5c). There are only few studies demonstrating Cerenkov photodynamic effect on cancer cell growth. Among them, cell growth arrest was related to the covalent link between the PS and the radionuclide or the concomitant treatment with nanoparticle doped with PS and addition of radionuclide. Kotagiri et al. [25] used TiO_2_ nanoparticles (as a photosensitizer) in association of ^18^F or ^64^Cu, against HT108 sarcoma cells grafted on nude mice. The benefit on tumour progression was however limited. Hartl et al. [24] assessed ^90^Y, as a β^−^ emitter, in the presence of P1 against murine glioma C6 and human breast MDA-MB-231 cells, in vitro, demonstrating cell growth inhibition. Duan et al. [21] used TiO_2_ nanoparticles in association with ^68^Ga-BSA or ^18^F-FDG against mouse breast cancer 4T1. Cell growth arrest was only found when cells were treated with nanoparticles in the presence of ^68^Ga-BSA, suggesting that the impact on cell growth is due to the relation between PS activation and energy produced during radionuclide decay as previously postulated [21]. Indeed, low treatment efficacy results were obtained with isotopes emitting low or modest particle energy (e.g., mean positron energy for ^18^F is 0.252 MeV, mean electron energy for ^90^Y is 0.935 MeV).

Herein, we demonstrated that X-ray radiations could be replaced by an unsealed source emitting β^+^ particles, namely, ^68^Ga. Clonogenic assay results obtained on U-251 MG cells with 3.0 MBq of ^68^Ga were comparable with previous study involving an X-PDT strategy at 2.0 Gy (Figure 5c) [29]. Hallmarks of X-rays consist of DNA damage, generation of oxygen derived species and, consequently, mitotic catastrophe and senescence, processes that are directly associated with radiotherapy [39]. In the case of PDT, DNA alteration could occur, while cells did not undergo cell death [40]. However, Duan et al. [21] showed that γ-H2AX level was increased when cells were treated with PS and ^68^Ga coupled to BSA but not with ^18^F-BSA, or when cells were exposed to each isotope alone, supporting the hypothesis that DNA damage in CR-PDT is associated with PS activation. Similarly, Yu et al. [41] demonstrated that the DNA repair mechanism is activated due to the presence of γ-H2AX foci when cells are exposed to both ^89^Zr isotope and PS. On the other hand, Paquot et al. [42] showed that mitotic catastrophe is a cumulative process with a peak occurring five days after irradiation exposure of human glioblastoma U-87 MG cells to gold nanoparticles. However, ^68^Ga irradiation at 1 MBq is a million times lower than that achieved by X-rays, supporting the concept that the impact on cell growth after cell exposure to ^68^Ga could only be due to the phototoxicity effect. On the basis of these results, it will be important to verify that the treatment with ^68^Ga is related to the mitotic catastrophe process over long periods of time. Finally, X-PDT relies on energy conversion from X-rays to visible light via the nanoscintillator. The visible light being, in turn, absorbed by the PS [5,6]. With β^+^ emitters, the nanoscintillating element can be excited by (i) the 511 keV annihilation photons and (ii) the UV part of the Cerenkov spectrum. The Cerenkov visible domain can directly activate the PS, the global system allowing three different activation ways. Whatever the initial energy of the photons, PS activation yields singlet oxygen production and, in addition, other radical species affecting the redox status of tumour cells [20,24,29].

In a recent analysis of the literature data, Klein et al. [33] concluded that there was no added value of photodynamic effect in X-PDT or CR-PDT. They used mathematical models to demonstrate that the added value in X-PDT was due to the presence of the nanoscintillator itself, which produces a radiosensitizing effect rather than a PDT effect itself. However, previous study demonstrated the absence of effective radiosensitizing effect of the nanoscintillator in X-PDT at energies between 160 and 320 kV using the same concentrations as the present study [29]. In addition, Klein et al. [33] attributed the CR-PDT efficacy to a potential combination of direct PS activation and radiosensitization. Our results are in contradiction with this assertion. Indeed, when we mixed P1 with ^68^Ga (Figure 1), we observed the production of singlet oxygen, which is the result of the type II photoreaction, highly specific to the PDT process. Moreover, exposing nanoparticles without PS to radioactivity did not induce any effect on the U-251 MG clonogenicity (Figure 5a), supporting the absence of radiosensitization at the radionuclide activity and lanthanides concentrations used in our experiments. In addition, we observed a significant PDT effect on cell clonogenicity at a ultra-low dose rate (roughly 10^−4^ mGy·MBq^−1^·s^−1^, data from OpenDose database) [43]. Thus, an efficient photodynamic effect can be achieved with very low dose exposure when the nanoscintillator/photosensitizer couple is correctly chosen. Furthermore, unlike PDT and X-PDT, CR-PDT could be applied to metastatic diseases thanks to the non-sealed nature of the light source.

In conclusion, Monte Carlo simulations demonstrated the added value of the presence of a nanoscintillator (as Tb) in AGuIX@ nanoparticles to increase singlet oxygen production when nanoparticles doped with photosensitizer are activated by Cerenkov radiation. This result has been confirmed experimentally (i) by singlet oxygen production assessment and (ii) cell survival decrease was demonstrated by clonogenic assays using glioblastoma cells pre-treated with AGuIX@Tb-P1 and exposed to ^68^Ga.

## 4. Materials and Methods

### 4.1. Reagents

Singlet Oxygen Sensor Green (SOSG) probe was purchased from Molecular Probe (Merck-Sigma, St Quentin Fallavier, France). Other reagents were of analytical grade.

### 4.2. Nanoparticle Solution Preparation

Ultra-small polysiloxane particles were synthesized as described previously [44]. Four different nanoparticles, based on the same AGuIX polysiloxane core surrounded by 1,4,7,10-tetraazacyclododecane-1,4,7,10-tetraacetic acid (DOTA)/metal cation (3+) complexes, covalently grafted to the inorganic matrix, were used. The cations embedded were Tb (Tb^3+^, Z = 65, A = 159 g mol^−1^) and Gd (Gd^3+^, Z = 64, A = 157 g mol^−1^). P1 was covalently grafted as a photosensitizer (AGuIX@Tb-P1 or AGuIX@Gd-P1). The procedure of synthesis and the characterization of the nanoplatform was described previously [29,44]. The AGuIX@ were suspended in water to obtain a stock solution of 3 mM P1 equivalent. The final Tb:P1 ratio was 16 moles Tb to 1 mole P1 for the AGuIX@Tb-P1. The concentration of AGuIX@Tb-P1 containing P1 will be referred to as the concentration of P1. Similarly, the AGuIX@Tb solution was 37.5 mM (Tb equivalent) in water. The AGuIX@Gd-P1 complexes were synthesized with a ratio of 1 mole of P1 to 25 moles of Gd, as estimated by inductively coupled plasma—mass spectroscopy analysis. The stock solution of AGuIX@Gd or AGuIX@Gd-P1 was, respectively, 50 mM Gd equivalent and 4 mM P1 equivalent in water.

### 4.3. Radiosynthesis of ^68^Ga-Citrate

The radiosynthesis was realized as previously described [45,46]. Briefly, [^68^Ga]-citrate was produced using mAIO module (TRASIS) by trapping of ^68^Ga-eluate obtained from a ^68^Ge/^68^Ga generator (around 400 MBq) on MCX cartridge and [^68^Ga]-citrate was eluted by 4 mL of a commercially available citrate solution (ACD formula A) towards a final vial. Approximatively 300 MBq of [^68^Ga]-citrate was produced with a radiosynthesis yield of 90.4% ± 1.0 decay corrected. Quality control analysis showed a radiochemical purity superior at 99% determined by radio-TLC and a pH value range of 5.5 to 7.0.

### 4.4. Singlet Oxygen Production with ^68^Ga Irradiation

The reaction mixture was prepared in 30 mM Tris/HCl (pH 7.4) containing 400 µM AGuIX@Tb-P1 or 45 mM AGuIX@Tb and 10 µM SOSG probe. Singlet oxygen quenching was achieved by addition of 10 mM sodium azide (NaN_3_ stock solution of 1 M) prepared in the same buffer. An additional experiment was performed with bovine serum albumin as a quencher [47] at a final concentration of 0.2% (*w*/*v*); the stock solution was 2% (*w*/*v*) in 30 mM Tris/HCl (pH 7.4) buffer. Irradiation was obtained by adding between 15 and 20 MBq of ^68^Ga. Fluorescence emission was detected spectroscopically at 525 nm for SOSG after excitation at 473 nm. An optical fibre was inserted in front of the vial containing the reaction mixture containing ^68^Ga to gather emission fluorescence photons. Emission spectra were recorded with a USB2000 spectrometer (Ocean Optics Inc, Dunedin, FL, USA). This versatile high-resolution spectrometer (FWHM = 3.5 nm) is an optical instrument based on a diffraction grating and a one-dimensional CCD detector array. Integration time was set to 5 s, the spectrum bandwidth ranged from 340 to 820 nm and the optical fibre was placed across from a transparent vial (Uvette 220–1600 nm; cat. No. 952010051, Eppendorf, Hamburg, Germany). Home-made software allowed long acquisition times and synchronization between laser illumination and signal recording. Integration time was set to 100 ms and time points were acquired every 5 min during 20 min.

### 4.5. Monte Carlo Simulations

#### 4.5.1. General Parameters

For the present study, we used Gate 9.1 with the Geant4 11.0.p01 and CLHEP 2.4.5 libraries. The pseudorandom generator was the Mersenne Twister. For the ionizing radiation physics, we chose the Livermore physics list and activated Rayleigh scattering, photoelectric, Compton scattering, pair conversion, electron ionization and Bremsstrahlung effects. Moreover, to assess optical physics, we added Cerenkov luminescence, as well as scintillation, Rayleigh and Mie scattering, optical absorption and fluorescence processes. Energy cut-off was set to 1 eV to follow up to P1 fluorescence wavelength.

#### 4.5.2. Geometry

Simulations involved a 10 mm side cubic world filled with water. This world contained spherical structures representing a U-251 MG cell (5 µm radius). Then clusters were defined inside the cell with a 0.5 µm radius. AGuIX@Tb/Gd-P1 were then modelled—to mimic the actual chemical properties of the nanoparticles: we defined a 5.5 nm radius silicone sphere bearing 18 Tb or Gd nuclei and one P1 molecule on its perimeter. The nanoparticle was repeated 6000 times into the cluster. Tb and Gd scintillations and absorption/emission characteristics were taken from the experimental measurements and added into a Gate material file [48].

#### 4.5.3. Source

The simulated source was designed as a 3 µm diameter sphere located outside the tumour cell. The emitted particles were positrons with an energy of 844 keV (i.e., mean ^68^Ga positron energy). To speed up the simulation, source was only emitted in the direction of the cell, the other directions being useless. For each experiment, 10^6^ primaries were tracked.

#### 4.5.4. Recorded Data

Phase space actors were placed at the cluster, scintillators and P1 edges to save the type of particle produced inside the cluster and nanoparticles as well as the type of incident particles (positron, scintillation, fluorescence or Cerenkov photons). In addition, we recorded the energy spectra inside the cluster for the different scenarios run. Data were analysed using ROOT 6.26/04.

#### 4.5.5. Simulation Scenarios

Four different scenarios were enacted to determine the interest of the presence of the nanoscintillators to active P1:(i)Nanoparticles were replaced by water to observe the raw Cerenkov spectrum.(ii)We used nanoparticles without Tb or Gd but with P1. That allowed the observation of direct PS activation through the Cerenkov visible part.(iii)We added Tb to the nanoparticle to observe scintillation and nanoscintillator fluorescence (due to Cerenkov UV photons) contributions.(iv)At last, we replaced Tb with Gd to validate the overall simulation results. Gd was not expected to transfer energy to the PS, and we expected to obtain a similar P1 fluorescence as in ii.

### 4.6. Biological Experiments

#### 4.6.1. Cell Culture

Human U-251 MG (ECACC 09063001, Salisbury, UK) glioblastoma-derived cells were cultivated in Roswell Park Memorial Institute medium (RPMI) without phenol red, containing 10% (*v*/*v*) heat-inactivated (30 min at 56 °C) foetal calf serum (Invitrogen, Paisley, UK), 1% (*v*/*v*) non-essential amino acid (Invitrogen), 0.5% (*v*/*v*) essential amino acid (Invitrogen), 1 mM sodium pyruvate (Invitrogen), 1% (*v*/*v*) vitamin (Invitrogen) 0.1 mg·mL^−1^ of L-serine, 0.02 mg·mL^−1^ L-asparagine (Merck-Sigma) and 1% (*v*/*v*) antibiotics (10,000 U·mL^−1^ penicillin, 10 mg·mL^−1^ streptomycin) (Merck-Sigma). The cells were seeded routinely at 10^5^ cells/mL and cultivated at 37 °C in a 5% CO_2_-humidified atmosphere (Incubator Binder, Tübingen, Germany).

#### 4.6.2. Anchorage-Dependent Clonogenic Assay

The clonogenic assay was performed in 6-well-plates with U-251 MG cells seeded at 500 cells/well. Cells were then treated in the presence of 16.6 µM AGuIX@Tb, 25 µM AGuIX@Gd, 1 µM AGuIX@Tb-P1 or 1 µM AGuIX@Gd-P1 (P1 equivalent concentration) for 24 h at 37 °C. After incubation, cells were washed with 2 mL of DPBS. Two millilitres of complete medium were added to each well before ^68^Ga addition. Activity concentration in each well varied from 0.05 to 1.5 MBq/mL. After complete ^68^Ga decay, cells were left to grow at 37 °C for 7 days. In parallel, cells were treated with 1 µM AGuIX@Tb-P1 (P1 equivalent concentration) for 24 h. Cell layers were washed with PBS and fresh medium was added to each well before X-ray exposure at 2.0 Gy, 320 kVp (X-ray Irradiator X-RAD 320-Precision X-ray INC., North Branford, CT, USA). Furthermore, cell clones were successively washed with 2 mL DPBS, fixed with 1 mL of 4% (*v*/*v*) formol (pH 7.4) at room temperature for 15 min, washed with 1 mL of DPBS and stained for 30 min with 0.05% (*w*/*v*) crystal violet solution prepared in DPBS and containing 25% (*v*/*v*) methanol. Finally, cells were washed three times with 2 mL of distilled water. Cell clones obtained were analysed after picture capture (GelCount, Oxford Optronix, Abingdon, UK) and ImageJ quantification (N.I.H., Bethesda, MA, USA) performed. Image analysis was performed with the well area taken as 862 mm^2^. Cell clone counting was improved by background subtraction. Data from untreated and treated cell conditions were compared and expressed as the mean ± SD (*n* = 12).

#### 4.6.3. Statistical Analysis

Results obtained were analysed using the Kruskal–Wallis test (α = 0.05) and post hoc Dunn’s test (α = 0.05) for paired groups. Any difference was considered significant at *p* < 0.05.

## Data Availability

Data is contained within the article and Appendix A.

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
