# Peer review of "Added Value of Scintillating Element in Cerenkov-Induced Photodynamic Therapy"

_pharmaceuticals, 2023, doi:10.3390/ph16020143_

Round 1

Reviewer 1 Report

The research article submitted by Schneller et al. is very interesting. It discusses the results from free 68Ga could substitute X-rays of X-PDT to obtain a cytotoxic phototherapeutic effect and improved effects due to the Tb energy transfer. The manuscript consists of thorough experimental evidence and has added a Monte CARLO simulation. Still, it lacks a few points: There are excellent strategies where nanoscales molecules or hybrids or nano lipids or nanohybrids. I would recommend using some references from Hasan’s and Kessel’s work in the introduction, such as: https://doi.org/10.1515/nanoph-2021-0191 , https://doi.org/10.1016/j.nantod.2020.101052

Since the generation of singlet oxygen is way lower in CR-PDT compared to conventional PDT. Would you prefer to compare the same with X-PDT?

Why CR-PDT would be an excellent method?

Why did you choose Tb instead of Eu to replace Gd, which could be a better emitter in near red (IR) than green emission and synergetic effect? Of course, PS agents should be different then.

SOSG fluorescent probe is okay to use, but usually, an absorbance probe is used to better understand the “numbers” due to multiple optical phenomena.

The manuscript requires a supplementary manuscript with relevant absorbance, emission, IR data (explanation of the formation of materials), and other relevant and control data to better understand this manuscript.

I would recommend the publication after a revision. 

Author Response

Thank you very much for your remarks.

Please find in attachment our point-by-point response. We modified the manuscript accordingly.

Best regards.

Reviewer 2 Report

In this work, Schneller and co-workers tried to explore the relevance of scintillating elements in Cerenkov-induced photodynamic therapy (CR-PDT). More specifically, the authors aimed to test whether free 68Ga can substitute X-rays in X-ray induced photodynamic therapy (X-PDT), in order to obtain a cytotoxic phototherapeutic effect. This analysis was performed by measuring singlet oxygen production, performing Monte Carlo simulations, and studying the survival of U-251 MG cells by clonogenic assays. The authors concluded that CR-PDT could indeed be an alternative to X-PDT.

This topic of research is indeed within the scope of this journal, and the conclusions of the present paper could be of interest for its readership. The paper is generally well-written. However, there are aspects that still require improvement and clarification. Thus, my recommendation is for Major Revision.

-Figures 1-4 can be difficult to analyze for colorblind readers. The authors should improve this;

-Abbreviations should be explained when first used. However, there are abbreviations (such as SOSG) that are used without explanation until the last section of the paper (4.). This should be corrected;

-Figure 2 requires error bars;

-Why just presenting R2 for AGuIX@Tb-P1 data (Figure 2)?

-Why there is only a zero point for 68Ga data (Figure 2)?

-The authors indicate that singlet oxygen production follows 68Ga decay (Figure 2). However, the slope of singlet oxygen production appears to be 1.25 times higher, and the evolution of these parameters over time do show noticeable differences. What is the relevant of these observed differences?

-Section 2.2 is very difficult to understand as the authors are not properly explaining what they are doing, how they are doing it and why they are doing it.

-A positive control should be added to the assays presented in section 2.3. Without it, it is very difficult to truly assess the impact of these systems on U-251 MG cell survival.

Author Response

(The authors gave the same response as above.)

Round 2

Reviewer 2 Report

The authors have addressed my comments, and so, my recommendation is for acceptance.